# Sensory Perception and Willingness to Pay for a Local Ancient Pear Variety: Evidence from In-Store Experiments in Italy

**DOI:** 10.3390/foods13010138

**Published:** 2023-12-30

**Authors:** Sergio Rivaroli, Massimiliano Calvia, Roberta Spadoni, Stefano Tartarini, Roberto Gregori, Cristina Calvo-Porral, Maurizio Canavari

**Affiliations:** 1Department of Agricultural and Food Sciences, Alma Mater Studiorum—Università di Bologna, 40127 Bologna, Italy; massimiliano.calvia2@unibo.it (M.C.); roberta.spadoni@unibo.it (R.S.); stefano.tartarini@unibo.it (S.T.); roberto.gregori4@unibo.it (R.G.); maurizio.canavari@unibo.it (M.C.); 2Business Department, Facultad Economía y Empresa, University of A Coruna, 15006 Coruna, Spain; ccalvo@udc.es

**Keywords:** just-about-right-scale, penalty analysis, willingness to pay, Becker–DeGroot–Marschak, local ancient pear, Italy

## Abstract

Product optimisation is one of the most crucial phases in the new product development or launch process. This work proposes applying penalty analysis to investigate the impact of not just-about-right (JAR) sensorial aspects on willingness to pay (WTP) and an overall liking for a local Italian ancient pear variety and to verify the mediating role of pleasantness in the relationship between not-JAR sensory attributes and consumers’ WTP using structural equation model (SEM). One hundred and twelve non-expert participants recruited during an in-store experiment evaluated overall liking and JAR attributes and were involved in an in-field experimental auction based on the non-hypothetical Becker–DeGroot–Marshak (BDM) mechanism. The participants’ average WTP for the sample was EUR 3.18 per kilogramme. Only juiciness and sourness significantly impact consumers’ overall liking but not on consumers’ WTP. Moreover, pleasantness did not mediate the relationship between non-balanced sensorial aspects and WTP. In conclusion, the penalty analysis for attributes not being JAR in monetary and hedonic terms is a beneficial research approach for a deep-inside evaluation of the potentiality of the product in the marketplace, providing helpful directions for product optimisation. Results show market potential for the local ancient pear variety ‘Angelica’.

## 1. Introduction

Consumers’ growing expectations for quality, healthiness, ecological and cultural content have aroused interest in local food products [1,2,3,4]. The local food movement has rediscovered some autochthonous genotypes of ancient local varieties of fruits and vegetables [5] recognised for their nutritional value and role in biodiversity conservation [6,7].

Most consumer goods industries strive to develop new products with high-added value to capture the attention of new customers and gain competitive advantage and long-term financial success [8]. Since the horticultural market is highly competitive, poorly differentiated, and characterised by low prices [9], ancient local fruits and vegetables have recently attracted retailers, breeders, and growers as cultivars that should meet consumers’ evolving needs and wants.

Launching a new fruit cultivar is a significant challenge. Knowing in advance whether a specific product is aligned with customers’ sensory expectations and individually perceived monetary values is crucial for proper strategic planning [10]. In general, variability in the perception of sensory attributes can influence the acceptability of food by consumers. In particular, the food industries need to verify whether the sensorial attributes of foods are optimal and to what extent variations in their characteristics could affect their liking and economic evaluation.

According to Nelson [11], consumers evaluate a product in two phases. First, they consider “search attributes”, that is, aspects that might be verified before the purchase (e.g., colour, size, and fruit shape). Second, they focus on “experience attributes”, which can be verified after the purchase and during the consumption experience (e.g., taste, juiciness, and crunchiness of fruit, among others). In other words, individuals’ appreciation for fruit consists of an intertwining of “search and experience” aspects that vary depending on the fruit that is going to be consumed [12,13,14]. Empirical literature considers experience attributes such as “compactness”, “juiciness”, “sweetness”, “sourness”, “aroma”, and “granulosity” to be relevant in determining consumer appreciation for fruits such as pears [15,16,17,18,19,20]. “Colour”, “shape”, and “size”, on the other hand, are those search quality traits validated by scholars, which affect the process of the acceptance of the fruit by consumers [19,21,22,23,24].

Many studies have focused on the chemosensory evaluation of fruit attributes [16,25,26,27]. However, an increasing number of research studies combine sensory and economic evaluations to offer a more exhaustive overview of how organoleptic properties of foods affect consumers’ intention to (re)purchase them. Examples of the latter across time and different consumer populations include Jaeger et al. [25], Pinto et al. [28], Rivaroli et al. [29], Costanigro et al. [30], McCluskey et al. [31], Karina Gallardo et al. [32], Gracia and Cantín [33], and Boccia et al. [34].

One way to assess a product’s optimal characteristics is by asking consumers whether they consider a sensory attribute too strong, weak, or “Just-About-Right” (JAR) [35]. A sensory attribute would not be at its ideal level should more than 20% of individuals rate it “out-of-JAR” [36]. For instance, Xiong and Meullenet [29] explored how much overall liking is affected when an attribute is not JAR, providing actionable information for optimising a product. Mean drop analyses (or penalty analysis) based on JAR data have also been used to establish which sensory attributes effectively impact consumer liking [31]. JAR scale and penalty analysis are widely employed in sensory and marketing studies to explore how consumers’ feelings toward a product combine with the hedonic scale to identify potential directions for product optimisation [37,38,39].

The consumers’ trade-off between quality and price can be explored regarding how sensorial aspects affect overall liking and value perception [8]. This is usually achieved in terms of willingness to pay (WTP), i.e., the maximum amount of money a consumer is willing to pay to buy a good. WTP can be studied using hypothetical and non-hypothetical value elicitation methods. Among non-hypothetical ones, experimental auctions have gained popularity among researchers over the past two decades due to their ability to mimic real market situations. Specifically, real products are exchanged for real money, thus allowing the WTP to be measured. According to Lusk and Shogren [40], this aspect is why non-hypothetical value elicitation methods can provide a more accurate and realistic value for the WTP. The Becker–DeGroot–Marshak (BDM) [41] auction mechanism, in particular, is very suited for conducting in-field auctions involving single participants [40]. As with other non-hypothetical value elicitation methods, the auctioned product, once presented, is usually used for a tasting experience.

According to Lawless et al. [42], integrating non-hypothetical valuation mechanisms and sensory applications can help understand how sensory attributes drive non-hypothetical WTP—or overall liking—thus improving product optimisation. From extant research on food pleasantness evaluated through the hedonic rating scale, it emerges that food’s liking is the predominant driver of WTP [43,44]. In contrast, much of the research testing the antecedents of overall liking using JAR questions is contradictory. For example, Prescott et al. [45] highlight that focusing on the sensory aspects of each product can inhibit the overall representation of consumers’ liking, inducing a hedonic bias. Conversely, among others, Jaeger et al. [46] reported a lack of this hedonic bias.

This paper contributes to the literature by looking at sensory-economic aspects linked to an ancient local pear genotype, namely the pear ‘Angelica’. It does this by jointly analysing the relationship between JAR data and overall consumer satisfaction—or the perceived monetary value—of goods, aspects often considered separately in the literature. In particular, considering the abovementioned debated role of using the product’s overall liking and specific sensory aspects in research, it considers the mediating role of pleasantness in the relationship between “out-of-JAR” sensory attributes and consumers’ perceived value expressed in terms of WTP.

Specifically, the study aims to explore the market potentiality and acceptability of the pear ‘Angelica’, a local ancient pear variety, by answering the following research questions:

(RQ1) Which “out-of-JAR” sensory attributes influence consumers’ overall liking and WTP more than others?

(RQ2) What is the consumer’s liking and WTP for the local ancient pear variety ‘Angelica’?

(RQ3) Does pleasantness significantly mediate the relationship between out-of-JAR sensory attributes and consumers’ economic appreciation?

The purpose of this study is, thus, two-fold. First, it identifies directions for future breeding programs and agronomic precautions by uncovering undesirable features of the ancient local pear cultivar ‘Angelica’ and their effects on consumers’ overall liking and WTP. Second, it improves the usefulness of combining sensory and BDM economic evaluation mechanisms. In this context, this study seeks to fill the gap left by the limited literature concerning the economic valuation of out-of-JAR sensory attributes, thereby providing helpful indications for product optimisation.

The remainder of the article is set up as follows. After explaining the study’s experimental design, the summary statistics of the data are presented. Next, the results are discussed, and the paper concludes by explaining the implications for the ancient local variety of pear investigated.

## 2. Materials and Methods

Five in-field studies were conducted from 29 October to 26 November 2022, with an average of 22 participants each. “In the field” experiments were chosen to reduce sample selection bias as participants were intercepted rather than self-selected [47]. Each participant underwent a “hedonic+JAR+BDM” product evaluation task lasting approximately 11 min (Table 1).

### 2.1. Participants

One hundred and twelve consumers were intercepted during five in-store sessions. The number of participants in each experiment ranged from 17 to 26. Participants were recruited during their shopping time and screened on eligibility criteria. Participation has been randomised. More specifically, only those passers-by who had not participated in previous studies were invited. All people involved in the organisation of the study were excluded from participation. People aged 18 years or older and who were residents in Italy were recruited based on three criteria: their interest in participating, responsibility for family food purchases, and the consumption of fruit at least once a week. Referring to this last recruiting criterion, rather than the consumption of local ancient pear, which is generally unknown by the vast majority of consumers, it is relevant that customers include fruit in their consumption habits to have consistent answers.

They gave online informed consent, and no cash incentives were used. Participants ranged in age from 19 to 74. The percentage of female participants was 62.50%. 65.18% consumed fruit daily, while 26.79% consumed fruit almost every day. The average rate of valid responses helpful for the analysis was 88.2%.

The study protocol was approved by the Ethics Committee at the Alma Mater Studiorum—Università di Bologna (Prot. n. 0,146,127 del 01/07/2022). Participants gave voluntary consent online and were assured that their responses would remain confidential. Each participant was free to leave the study at any time and without justification.

### 2.2. Sample

The pear ‘Angelica’ (Pyrus communis) is an old variety of pear of unknown origin in the Italian germplasm that was selected among four other ancient varieties of pear to verify its attractiveness and market potentiality. The first documents describing this variety date back to the XVI century and consist of the botanical tables of Ulisse Aldrovandi [48]. The pear ‘Angelica’ is represented in Bartolomeo Bimbi’s paintings [49] from the XVII century. The pear ‘Angelica’ is a very productive variety. It is susceptible to the main pathogens and pests. Its fruits are usually harvested at the end of September and are medium-sized, with smooth yellow skin, juicy, and aromatic. The fruits’ shelf life is short; for this reason, they must be consumed shortly after harvest. Their cultivation is now limited to a few commercial orchards in the Italian regions of Emilia-Romagna, Veneto, and Marche.

The pear ‘Angelica’ samples employed throughout this study were harvested on August 31st in a commercial orchard in Bagnacavallo (RA, Italy) and immediately stored at the standard temperature of 4–5 °C. The fruits were removed from the refrigerator 24 h before the tasting sessions. Only those fruits without visual defects were utilised. They were cleaned, placed in a resealable food box and offered to interviewees for the follow-up phase of the study. Once stored, a sample of 20 fruits was used to evaluate some fruit quality standards. Firmness was measured using a penetrometer (8 mm diameter probe) on the opposite sides of pear surfaces. Soluble Solids Content (SSC) was determined using a digital refractometer (Atago, Tokyo, Japan) on filtrated pear juice. Titratable acidity (TA) was detected with an automatic titrator (Crison Instruments, SA, Barcelona, Spain). Twenty millilitres of juice diluted with an additional twenty millilitres of distilled water were titrated to pH 8.1 with 0.25 N NaOH. The quality analysis of the fruit showed an average firmness of 3.74 kg/cm^2^, a soluble solid content of 15.7 °Brix, and a titratable acidity of 1.69 g/L of malic acid.

### 2.3. Procedure

A vending table with three tablets was set up in the store to conduct more experiments simultaneously. A second vending table was organised with the pear samples to be auctioned. All sessions were arranged according to the protocol and followed “JAR + Hedonic + BDM” tasks. In other words, each participant in the experiment underwent a sensory test, answered the JAR questions, provided hedonic scores, and participated in the BDM auction.

Figure 1 shows the flowchart of each auction session. In Step 1, participants were informed about the nature of the study, its benefits, and its risks. They were informed that the research lasted approximately 15–20 min, and there was no compensation for participating. In Step 2, answers were collected concerning primary demographic data (e.g., genre, age) and other personal information (e.g., responsibility for family food purchases, price usually paid for one kilo of pears, fruit consumption frequency). Some questions were organised such that their answer followed a binary category., i.e., “yes” or “not”, whereas others were according to a frequency category, i.e., “never”, “less than once a week”, “once or twice a week”, “almost every day”, and “every day”. In Step 3, each participant was first given a pear sample. Before any sensory evaluation, they were served water and a neutral biscuit (e.g., non-salted crackers) to cleanse their palate. Participants were asked to describe the sample in terms of 7 variables, namely “dimension”, “compactness”, “juiciness”, “sweetness”, “sourness”, “aroma”, and “granulosity”, using a 5-point JAR scale (1 = “Not enough…”, …, 3 = “Just about” right, …, 5 = “Too much...”). Furthermore, participants were requested to provide an overall liking of the sample on a scale from 1 to 9, where 1 was “strongly dislike”. In line with Xiong and Meullenet [39], each JAR variable was split and re-coded into two dummy, i.e., binary, variables, namely “too low” and “too much”, taking advantage of the fact that the central category of the 5-point JAR scale represents the ideal level of the sensory attribute. Variable “too low” maps responses 1 and 2 to 1 and the others to 0. The variable “too much” maps responses 3 and 4 to 1 and the others to 0. In other words, score 3, i.e., the ideal level, always maps to 0 independent of the dummy considered. This procedure translates into 14 binary JAR-recoded variables, i.e., 7 pairs. In Step 4, consumers were introduced to the BDM mechanisms through a non-bidding auction round. In Step 5, the participants were informed through the instructions on the tablet that they would have participated in the actual BDM auction. Specifically, each participant was asked about their maximum WTP for one kilo of pear ‘Angelica’. Each bid was compared with a price randomly drawn by the computer from a uniform distribution ranging from EUR 1.50 to EUR 5.00 per kilo. Participants were not aware of the price distribution. In Step 6, consumers received the auction result in which they participated. They won the auction if their bid was higher than the random price; they lost the auction if the bid was lower. In the final step, the results are translated into real outcomes. In case of victory, they were invited to choose one box containing one kilo of pear ‘Angelica’ and pay the randomly generated price. In case of loss, they did not incur any expenses and left the place empty-handed. Data were collected via a survey on Qualtrics^®^ administered on tablets.

### 2.4. Data Analysis

Data analysis was performed using STATA 17.0 (StataCorp, LLC., College Station, TX, USA) according to the following steps. First, the empirical demand curve was computed to show how pear ‘Angelica’ quantities vary across sample prices. The revenue as a function of product price was also added to find the optimal price, maximising revenues.

Second, a penalty analysis was conducted, which combined information from JAR scales and overall liking data to identify which attributes determine a drop in a product’s pleasantness. Specifically, participant responses were used to calculate, for each attribute, the difference between the mean liking scores of those who reported an attribute being JAR and those who rated the product “out-of-JAR” (“too low” and “too high”, taken together). The average drop in the overall liking score was plotted against the percentage of consumers who reported a deviation of a specific attribute from the JAR. This helped identify those attributes which could be considered “out-of-JAR”. As Xiong and Meullenet [29] pointed out, only those attributes that deviated from the ideal for at least 20% of consumers were considered “out-of-JAR”.

Finally, a Sobel test [44] via the structural equation model (SEM) was carried out to determine whether the consumer’s overall liking mediates the effect of the binary “out-of-JAR” attributes on the WTP (Figure 2). SEM is a type of multivariate analysis that combines factor and path analysis. SEM focused exclusively on path analysis because no latent variables are involved in this study. The structural model, expressed in terms of the relationships between variables, was estimated and tested to evaluate the theoretical links [50]. The adjusted Baron and Kenny approach [45] was used to test the mediating effect, which consists of four steps. First, the direct relationship between independent and moderating variables (X → M) is assessed. Second, the direct relation between mediating and dependent variables (M → Y) is estimated. Third, the direct relation between independent and dependent variables (X → Y) is measured. Last, the significance of the Sobel test (z-value ≥ 1.96, *p* ≤ 0.05) is checked. This approach may potentially result in three different outcomes. First, should one of the X → M and M → Y relations be not significant, no mediation effect would exist. Second, a complete mediation would occur if Sobel’s z-test is significant, but the direct relation between independent and dependent variables (X → Y) is not. Finally, partial mediation occurs when all four steps above are significant. Emphasis is put on how consumers’ overall liking indirectly mediates between “out-of-JAR” dummies variables and fruit intake (1 = “Never”; 5 = “Every day”) and consumer WTP. This is estimated based on the direct effect of the binary “out-of-JAR” variables and the fruit intake variable on the mediator variable (Path A, i.e., X → M), on the one hand, and the direct effect of the mediator variable on the dependent variable WTP (Path C; i.e., M → Y), on the other hand. Calculations are carried out employing the *medsem* package [51]. The model also includes regression paths between consumers’ knowledge of average pear prices (How much does a kilo of pears usually cost in EUR/kilo when you shop for them?) and consumers’ WTP to detect the robustness of WTP judgements (Path D). The goodness of fit of the SEM model has been evaluated using the cutoff criteria of the root mean square error of approximation (RMSEA; values close to 0.08 or below), the standardised root mean square residual (SRMR; values close to 0.08 or below), the comparative fit index (CFI; values above 0.95), the normalised chi-square (CMIN/DF; values close to 5 or below), and the standardised loading estimates (0.30 and above) [52,53]. Convergence validity was tested by looking at the relative average variance extracted (AVE; close to or above 0.05). The structural model was evaluated in terms of the coefficient of determination (*R*^2^; 0.25 = weak, 0.50 = moderate, 0.75 = substantial) and path coefficients (between −1 and +1). A bootstrapping procedure with 1000 sub-samples was performed to evaluate the significance of path coefficients.

## 3. Results

### 3.1. Overall Liking and Economic Assessment

The product’s pleasantness and economic assessment reveal high consumer acceptance for pear ‘Angelica’. The average overall liking score for the pear sample was 7.78 (SD = 1.16). The average WTP emerging from the BDM mechanism was EUR 3.18 per kilo (SD = 1.21). This value is higher than the average level of consumer awareness regarding the price of pears (M = EUR 2.90 per kilo; SD = 0.79).

Figure 3 presents the empirical demand curve emerging from the experiment, which captures the relationship between price and quantity of pear ‘Angelica’. The empirical demand for pear ‘Angelica’ is analysed using a “constant elasticity model” [54], i.e., a log-log model estimated through OLS as witnessed by Equation 1, where natural logarithms are employed.
(1)ln (Quantity)=α+β×ln (Price)

In particular, parameter *β*, being defined as (%∆ *Quantity*)/(%∆ *Price*) [54], clearly represents an elasticity. The relationship comprises 35 observations, i.e., 35 unique prices for 35 pear Angelica quantities. A total 35 observations out of 112 were obtained because of multiple equal price values. The estimates are $\hat{\alpha }=5.40\;(p<0.01)$ and $\hat{\beta }=-1.60\;(p<0.01)$ and significant. It is worth emphasising that the elasticity of the demand curve concerning price is equal to −1.60. The sign reflects the downward-sloping behaviour typical of demand curves. More precisely, a 1% increase in the pear ‘Angelica’ price reduces the quantity consumed by 1.60%.

Figure 4 shows the potential revenue linked to different price levels. The optimal price (i.e., dotted line) that maximises the revenue equals EUR 3.00 per kilo.

### 3.2. Penalty Analysis

Figure 5 shows the average drops in overall liking as a function of the proportion of consumers that selected the option “out-of-JAR” to describe each of the attributes of the pear sample, thus indicating a need for product improvement. Specifically, in the penalty analysis, respondents are grouped in the “Not-To-JAR” (i.e., “Too little” or “Too much”) or JAR group. Then, the percentage of consumers in “Not-To-JAR” is calculated, and corresponding mean liking scores for the “Not-To-JAR” and JAR categories are estimated. Results are thus represented graphically by plotting the penalty (i.e., the mean drops of liking) against the percentage of consumers “Not-To-JAR”. A penalty is usually not considered if the percentage of consumers in the “Not-To-JAR” group is less than 20%. In this study, four attributes scored the highest deviation from the ideal level, i.e., mean drop. These are the small calibre of the fruits (“−Dimension”), the too-low juiciness (“−juiciness”), the too-low sourness (“−sourness“), and the inconspicuous aroma (“−aroma”). These aspects were thus considered in the path analysis using SEM.

### 3.3. Structural Model Evaluation and Path Analysis

Before testing the structural relationships of the structural model, its validity was evaluated by examining those recommended critical indexes mentioned above. A CFI of 0.993 (≥0.95), an RMSEA of 0.071 (≤0.08), an SRMR of 0.019 (≤0.08), and normalised Chi-square values (i.e., ≤5) suggest a good fit of the structural model. The coefficient of determination (*R*^2^), which explains the model’s predictive accuracy, is equal to 0.52, indicating that the proposed models have moderate explanatory power. Results on direct (standardised) path coefficients are presented in Table 2. They show that perceiving less juiciness (*β* = −0.417; *p* = 0.000) and less sourness (*β* = −0.200; *p* = 0.032) negatively impacts overall liking. Instead, the relationship between P_0_ (perceived general value of common pears sold in the market) and WTP is statistically significant and positive (*β* = 0.566; *p* = 0.000).

### 3.4. Mediating Effect

Baron and Kenny’s approach was used to statistically ascertain the mediating effect of consumers’ overall liking. Along Path A (X → M), it emerges that the direct relationships between the attributes “too less juiciness” and “too less sourness” and the mediating variable “overall liking” are, respectively, the only significant ones. The relation between “overall liking” and WTP, i.e., the dependent variable, is not significant along Path C (M → Y). Therefore, no significant indirect mediation effect (X → M → Y) occurs between predictors and WTP (Table 3). All z-scores of the Sobel test are not statistically significant (*p*> 0.10). The values are mainly below criterion Z > 1.96, indicating no statistically significant overall liking mediating effect.

## 4. Discussion

Prior works on food acceptability focus separately on the impacts of product sensory attributes on either WTP or overall liking using JAR sensory evaluation and non-hypothetical valuation tasks [42].

Consumers’ acceptability for the ancient local pear variety ‘Angelica’ has been estimated using WTP elicitation through “JAR+Hedonic+BDM” tasks. While previous studies suggest that food liking remains the primary driver of WTP [35,42], the mediating role of overall liking in the relationship between JAR sensory attributes and their economic appreciation remained underexplored. Focusing on suboptimal sensory aspects has provided a valuable direction for product optimisation. The use of penalty analysis—to select “out-of-JAR” attributes—and SEM allowed us to test the extent to which consumers’ overall liking mediates the effect of these attributes on the WTP.

This study finds that the ancient local variety of pear ‘Angelica’ was particularly appreciated by consumers both in hedonic and economic terms (Research Question 2). Penalty analysis findings reinforced this result, considering the restricted number of sensorial aspects perceived “out-of-JAR”. Results showed room for improvement only for two of four unbalanced sensory aspects identified by the penalty analysis: juiciness and sourness (Research Question 1).

The analysis showed that overall liking for this fruit does not influence consumers’ WTP or mediate between “out-of-JAR” attributes and WTP (Research Question 3). The only variable influencing consumers’ economic appreciation is the reference price they have in mind for other marketed pear varieties they have previously experienced. It also emerges that the frequency of fruit consumption does not increase WTP or overall liking.

## 5. Conclusions and Implications

### 5.1. Conclusions

Within the research on product optimisation, there are limited results in the literature concerning the economic evaluation of out-of-JAR sensory attributes. Hence, this study seeks to fill the gap and thus can be considered novel.

The study highlights the critical drivers for planning appropriate marketing strategies to enhance the Angelica cultivar. It identifies which key aspects could be considered for planning future breeding programs and which agronomics precautions could be adopted to minimise the undesirable features of this local ancient pear variety. More specifically, by increasing the juiciness and sourness of the fruit, it would be possible to offer consumers a positive and memorable tasting experience. Sourness (or acidity) is a fundamental trait for fruit sensory intensity that affects the acceptability and consumption of fruit [55]. Juiciness (i.e., the amount of liquid released during mastication), a relevant index of pear quality and freshness in describing the ideal pear [20], is directly correlated with sourness and sweetness. Although juiciness and sourness are two aspects that deviate from what customers expect to experience during the tasting experience, a well-balanced mix of all other sensory aspects largely compensates for this disadvantage, leading to an excellent hedonic and economic valuation.

### 5.2. Academic Implications

Some implications follow empirical results. One of the study’s main contributions is to shed light on whether the sensorial congruence goes in the same direction as economic and hedonic evaluations of the product, thus enriching the extant literature of sensory research. Although a disjoint impact of product sensory attributes on overall liking or WTP is prevalent in consumer behaviour [42,44,56,57,58], few studies have tested how consumers’ overall liking mediates between specific sensorial attributes and consumers’ economic evaluation (e.g., WTP) of a product. Results suggest that incongruent sensorial aspects decrease the economic valuation of a product based on consumers’ natural unpleasantness, although this relation appears not significant (Research Question 1). Pleasantness, the hedonic value of affective response to sensorial stimuli received by consumers (e.g., taste), has been widely accepted as a significant predictor of consumers’ WTP [42,44,56,57,58]. Furthermore, research shows that pleasantness mediates the link between the sensorial aspects of a product experience and consumer behaviour [59]. By contrast, the results of this work have not proven that pleasantness mediates between “out-of-JAR” sensory attributes and the economic evaluation of the product (Research Question 3). This could be because pear ‘Angelica’ is an ancient local variety not widely known to participants.

This research also tries to contribute to the literature on penalty analysis. In particular, it suggests that no significant direct impact on consumers’ economic product evaluations happens when the mean drop in the overall liking score of unbalanced sensorial aspects is not evident and when only a few sensorial aspects are considered “out-of-JAR”. In line with Prescott et al. [45], this result is likely because consumers, during the personal process of food evaluation, express a global hedonic judgment based on the product as a whole rather than on specific aspects.

With respect to the extant literature, empirical results display a different effect of overall liking towards consumers’ WTP. For example, Seppä et al. [44] show that pleasantness predicts consumers’ WTP for apples. According to Lawless et al. [42], an overall liking score of seven or above (measured using a nine-point verbal hedonic scale) is a significant and positive predictor of WTP for new nutraceutical-rich juice products. Again, using two independent PLSR models, the authors evidenced that attributes considered not optimal (not JAR) elicited the same effect on overall liking and WTP. On the contrary, our findings show that the market price is the primary driver of consumers’ economic appreciation. In fact, while the buying process depends on emotional aspects, the process which generates the WTP is mainly cognitive-based, grounded on comparisons with prices of products of the same kind, i.e., pears in this case [60]. This sort of anchoring effect occurs because participants without previous experience with the product decide to bid based on the most effortlessly accessible value they know, e.g., the price of the pears they have previously consumed. Furthermore, as pointed out by Mussweiler and Strack [61], the intensity of this effect depends on the decision maker’s familiarity with the target object.

Finally, our study further confirms that consumers pay more for products they frequently consume, mainly because they know what they are paying for [44,62]. Our results show that the frequency of fruit consumption does not significantly affect pleasantness and WTP. This result can be explained because pear ‘Angelica’ is an ancient cultivar, relatively unknown to consumers.

### 5.3. Practical Implications

Maximising consumer satisfaction and future purchases is rooted in the knowledge of consumer preference and WTP for a product. The conclusions of this study lead to some managerial implications.

First, marketers, producers, and breeders should be aware of the congruence between sensorial aspects and overall liking when defining where to concentrate their efforts. Understanding the cost of suboptimal sensory attributes may provide helpful information to identify optimal strategies. Based on our results on the ancient local pear variety ‘Angelica’, offering a more juicy and acidic fruit will be relevant to gain customer loyalty and strengthen their economic appreciation.

Second, since consumers’ WTP revolves around the optimal price that maximises revenues, all actors in the food chain must better evaluate whether it is worth having all sensorial aspects in equilibrium towards an ideal product or to adopt good communication strategies to enhance intrinsic and extrinsic aspects that differentiate the product from other pears. In marketing practices, for example, retailers add desirable elements to arouse pleasant emotions in consumers and mitigate the effects of unbalanced sensorial aspects, thus increasing consumers’ emotional states.

Third, a good labelling strategy informing consumers that the pear is “ancient” and “local”, together with alternative forms of market channels such as farmer markets, enhancing direct contact between farmers and customers, would enhance the sensorial peculiarities of the product, thus contributing to building customer loyalty and raising WTP.

### 5.4. Limitations and Further Research

The approach adopted in this study provides useful information on consumer preferences for pear ‘Angelica’, an ancient local variety of pear. Future research, however, could extend the main findings of this work, addressing some of its limitations. One crucial research avenue is to use a larger random sample, which can help reduce the risk of selection bias and improve the findings’ reliability. Furthermore, consumer test results could be paired with a sensory analysis based on a quantitative descriptive analysis and a panel of experts to obtain a detailed sensory profile of the ancient local pear cultivar.

Despite the abovementioned limitations, we believe this interdisciplinary study could help raise awareness among plant breeders and marketing managers to jointly consider sensorial and economic aspects to improve the competitiveness of ancient local varieties such as the pear ‘Angelica’.

## Figures and Tables

**Figure 1 foods-13-00138-f001:**
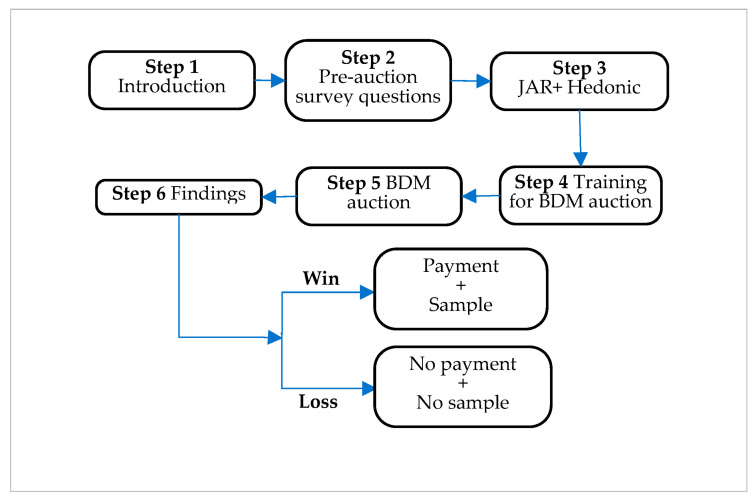
Flowchart of the auction sessions.

**Figure 2 foods-13-00138-f002:**
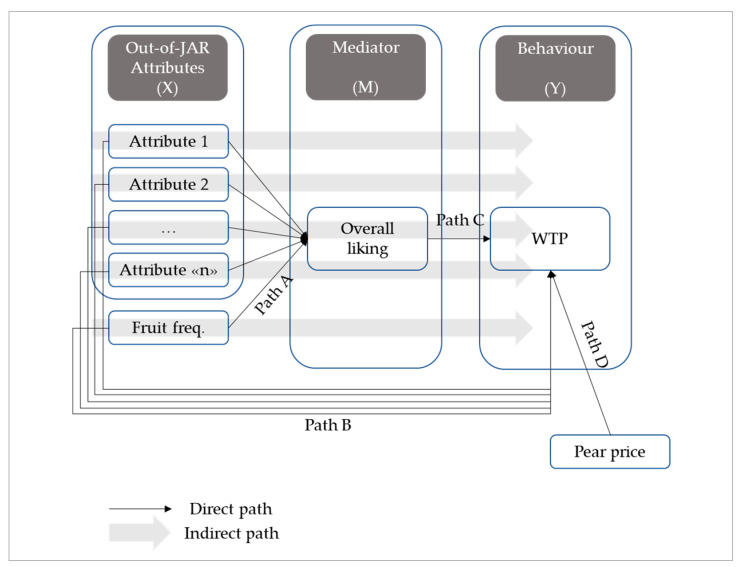
Representation of direct and indirect effects of attributes “out-of-JAR” on WTP.

**Figure 3 foods-13-00138-f003:**
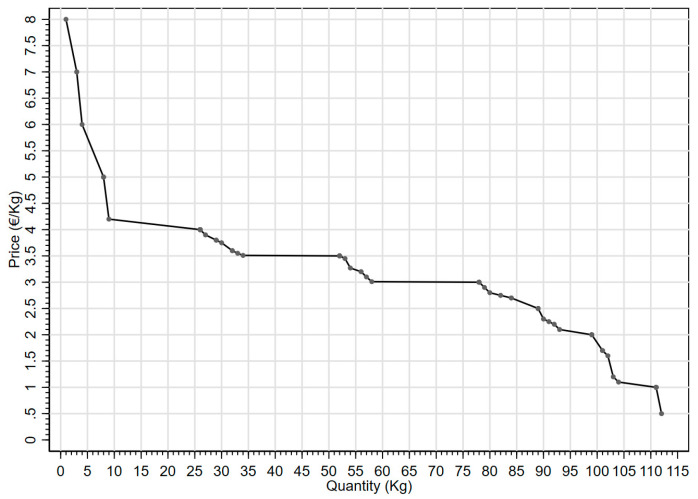
Empirical demand curve.

**Figure 4 foods-13-00138-f004:**
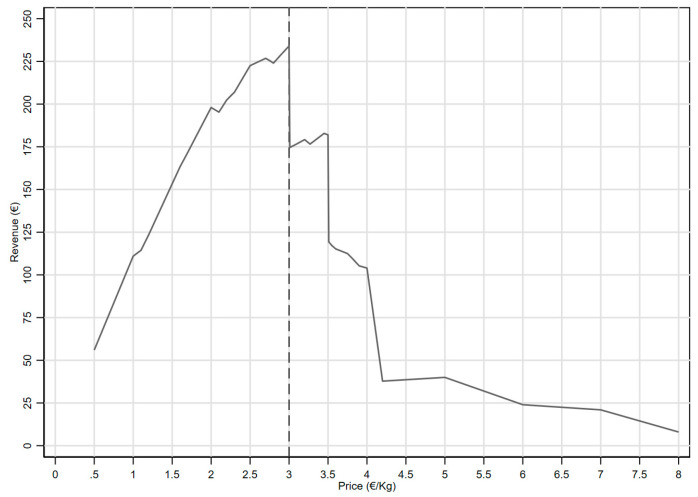
Revenue trend as a function of the product price.

**Figure 5 foods-13-00138-f005:**
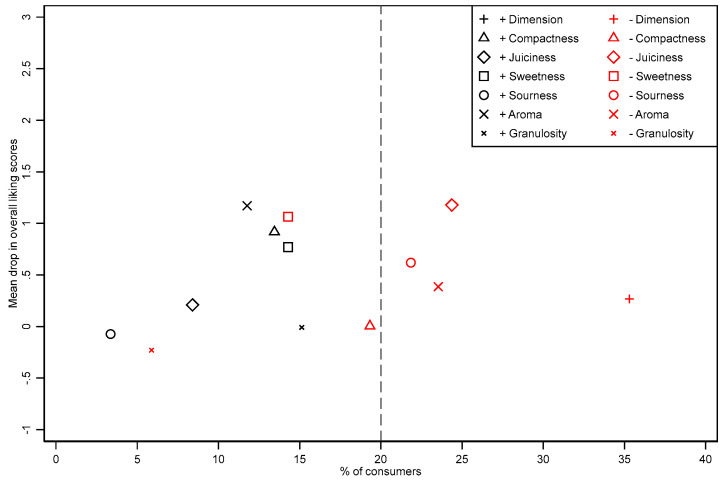
Mean drops in overall liking as a function of the percentage of consumers that checked an attribute “out-of-JAR”. Legend: −Dimension= ”too less calibre”, +Dimension= ”too much calibre”, −Compactness = “too less compact”, +Compactness = “too much compact, −Juiciness = “too less juicy”, +Juiciness = “too much juicy”, −Swetness = “too less sweet”, +Swetness = “too much sweet”, −Sourness = “too less sour”, +Sourness = “too much sour”, −Aroma = “too less aroma”, −Aroma = “too much aroma”, −Granulosity = “too less granulose”, +Granulosity = “too much granulose”.

**Table 1 foods-13-00138-t001:** Overview of in-field experiments.

Date	Participants	%	Average Duration (Minutes per Participant)
28 October 2022	19	16.96	16
3 November 2022	26	23.21	10
10 November 2022	25	22.32	9
17 November 2022	25	22.32	14
24 November 2022	17	15.18	9
Total	112		11

Notes. Experimental treatments: Hedonic + JAR + BDM; duration in minutes per participant.

**Table 2 foods-13-00138-t002:** Estimated regression coefficients included in the path model.

Path	Std. *β*.	SE	*p*-Value
▪ *Direct path*			
▪ *Path A (X → M)*			
▪ −Dimension → Overall liking	−0.072	0.090	0.420
▪ **−Juiciness → Overall liking**	**−0** **.417**	**0** **.076**	**0** **.000 *****
▪ −Aroma → Overall liking	-0.053	0.089	0.552
▪ **−Sourness → Overall liking**	**−0** **.200**	**0** **.094**	**0** **.032 ****
▪ Fruit intake → Overall liking	0.028	0.067	0.673
▪ *Path B (X → Y)*			
▪ −Dimension → WTP	−0.055	0.074	0.456
▪ −Juiciness → WTP	−0.031	0.100	0.753
▪ −Aroma → WTP	−0.107	0.075	0.157
▪ −Sourness → WTP	−0.054	0.081	0.505
▪ Food intake → WTP	−0.029	0.071	0.684
▪ *Path C (M → Y)*			
▪ Overall liking → WTP	0.124	0.103	0.231
▪ *Path D*			
▪ **P_0_ → WTP**	**0** **.566**	**0** **.080**	**0.00 *****

**Notes.** ** *p* < 0.05, *** *p* < 0.01. **Legend:** −Dimension= ”too less calibre”, −Juiciness = “too less juicy”, −Sourness = “too less sour”, −Aroma = “too less aromatic”, P_0_ = pear price, WTP = Willingness to pay.

**Table 3 foods-13-00138-t003:** Indirect (mediation) effects of predictors on WTP.

Path	Std. *β*.	SE	*p*-Value	Sobel(Z)
▪ −Dimension → Overall liking → WTP	−0.022	0.029	0.462	−0.750
▪ −Juiciness → Overall liking → WTP	−0.140	0.115	0.222	−1.400
▪ −Aroma → Overall liking → WTP	−0.018	0.034	0.597	−0.0589
▪ −Sourness → Overall liking → WTP	−0.070	0.060	0.246	−1.252
▪ Fruit intake → Overall liking → WTP	0.001	0.014	0.657	0.333

Legend: −Dimension= ”too less calibre”, −Juiciness = “too less juicy”, −Sourness = “too less sour”, −Aroma = “too less aromatic”, P_0_ = pear price, WTP = willingness to pay.

## Data Availability

Data is contained within the article.

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
