# Peer review of "Sensory Perception and Willingness to Pay for a Local Ancient Pear Variety: Evidence from In-Store Experiments in Italy"

_foods, 2023, doi:10.3390/foods13010138_

Round 1

Reviewer 1 Report

Comments and Suggestions for Authors

Overall, the writing of this article is neat and clear. A good discussion structure has provided a clear understanding of the scope of the study/article, and research gaps are explained by showing relevance to the contribution of the study. The title reflects the content of the manuscript well.

However, adding how the study operationalizes the sensory-economic aspects would help the readers understand the scope better. Furthermore, please describe in what way can the contributions be seen.

What is the research issue on this moderating variable of pleasantness?

Methodology - In terms of the participant’s requirement (screening question #3 - consumption of fruit at least once a week), the concern is - any fruit? Why? Why not specifically on local Italian ancient pear? Please justify this pre-determined criterion.

Why had the pear ‘Angelica’ (Pyrus communis) been the focus of this study? Any research and practical reasons?

In terms of data analysis, the discussion made about the results of the study including reliability and validity issues as well as the presentation of the analysis results have been well made and organized which facilitates reading and research on them.

The discussion and the remaining sections are also well-made and are clearly linked to the results of the study

Quality of communication - Overall, this manuscript is easy to read and interesting.

References - More recent articles need to be cited.

Author Response

Reviewer #1

Overall, the writing of this article is neat and clear. A good discussion structure has provided a clear understanding of the scope of the study/article, and research gaps are explained by showing relevance to the contribution of the study. The title reflects the content of the manuscript well.

            Thank you very much.

However, adding how the study operationalises the sensory-economic aspects would help the readers understand the scope better. Furthermore, please describe in what way can the contributions be seen. What is the research issue on this moderating variable of pleasantness?

Thank you for these comments. On page 3, lines 101-108, the sentences that clarify the Reviewer’s questions were modified as follows:

This paper contributes to the literature by looking at sensory-economic aspects linked to an ancient local pear genotype, namely pear ‘Angelica’. It does this by jointly analysing the relationship between JAR data and overall consumer satisfaction – or the perceived monetary value – of goods, aspects often considered separately in the literature. In particular, considering the abovementioned debated role of using the product’s overall liking and specific sensory aspects in research, it considers the mediating role of pleasantness in the relationship between “out-of-JAR” sensory attributes and consumers’ perceived value expressed in terms of WTP.

Methodology - In terms of the participant’s requirement (screening question #3 - consumption of fruit at least once a week), the concern is - any fruit? Why? Why not specifically on local Italian ancient pear? Please justify this pre-determined criterion.

Thank you for this comment. On page 4, lines 145-150, the sentence was modified as follows:

People aged 18 years or older and residents in Italy were recruited based on three criteria: their interest in participating, responsibility for family food purchases, and consumption of fruit at least once a week. Referring to this last recruiting criterion, rather than the consumption of local ancient pear, which is generally unknown by the vast majority of consumers, it is relevant that customers include fruit in their consumption habits to have consistent answers.

Why had the pear ‘Angelica’ (Pyrus communis) been the focus of this study? Any research and practical reasons?

Thank you for this comment. We inserted the motivation on page 4, lines 160-162, as follows:

The pear ‘Angelica’ (Pyrus communis) is an old variety of pear of unknown origin in the Italian germplasm that was selected among four other ancient varieties of pear to verify its attractiveness and market potentiality.

In terms of data analysis, the discussion made about the results of the study including reliability and validity issues as well as the presentation of the analysis results have been well made and organised which facilitates reading and research on them. The discussion and the remaining sections are also well-made and are clearly linked to the results of the study. Quality of communication - Overall, this manuscript is easy to read and interesting.

            Thank you very much. We really appreciate this.

References - More recent articles need to be cited.

Appreciating the reviewer’s comment, we have updated the literature in the introduction.

Added Bibliography:

1 Cappelli, L.; D’ascenzo, F.; Ruggieri, R.; Gorelova, I. Is Buying Local Food a Sustainable Practice? A Scoping Review of Consumers’ Preference for Local Food. Sustain. 2022, 14, 1–17, doi:10.3390/su14020772.

13 Roosen, J.; Neubig, C.M.; Staudigel, M.; Agovi, H. Product Appeal, Sensory Perception and Consumer Demand. Eur. Rev. Agric. Econ. 2023, 50, 1338–1363, doi:10.1093/erae/jbad020.

14 Cardona, M.; Izquierdo, D.; Barat, J.M.; Fernández-Segovia, I. Intrinsic and Extrinsic Attributes That Influence Choice of Meat and Meat Products: Techniques Used in Their Identification. Eur. Food Res. Technol. 2023, 249, 2485–2514, doi:10.1007/s00217-023-04301-1.

2 da Costa Marques, S.C.; Mauad, J.R.C.; de Faria Domingues, C.H.; Borges, J.A.R.; da Silva, J.R. The Importance of Local Food Products Attributes in Brazil Consumer’s Preferences. Futur. Foods 2022, 5, 100125, doi:https://doi.org/10.1016/j.fufo.2022.100125.

33 Gracia, A.; Cantín, C.M. Effects of Consumers’ Sensory Attributes Perception on Their Willingness to Pay for Apple Cultivars Grown at Different Altitudes: Are They Different? Foods 2022, 11, doi:10.3390/foods11193022.

34 Boccia, F.; Alvino, L.; Covino, D. This Is Not My Jam: An Italian Choice Experiment on the Influence of Typical Product Attributes on Consumers’ Willingness to Pay. Nutr. Food Sci. 2023, doi:10.1108/NFS-04-2023-0076.

38 Kamal Gosh, U.D.; Dey, M.M. Consumers’ Willingness-to-Pay for Newly Developed U.S. Farm-Raised Convenient Catfish Products: A Consumer-Based Survey Study. Aquac. Econ. \& Manag. 2022, 26, 332–358, doi:10.1080/13657305.2022.2060374.

Reviewer 2 Report

Comments and Suggestions for Authors

Thank you for the opportunity to read the text.
Below are my comments:
The goal may not be to answer the listing of 3 research questions.
The research question is for the purpose of achieving the stated goal. This paragraph needs to be worded differently.
You cannot end the chapter with a table (applies to the chapter presenting the results).
Please separate the discussion and summary into separate chapters.
I wish you to continue your fruitful work.

Comments on the Quality of English Language

The language is fine, some little mistakes.

Author Response

Reviewer #2

Thank you for the opportunity to read the text. Below are my comments:

The goal may not be to answer the listing of 3 research questions. The research question is for the purpose of achieving the stated goal. This paragraph needs to be worded differently.

Appreciating the reviewer’s comment, we have updated the sentence on page 3, lines 109-110 as follows:

Specifically, the study aims to explore the market potentiality and acceptability of the pear ‘Angelica’, a local ancient pear variety, by answering the following research questions:

You cannot end the chapter with a table (applies to the chapter presenting the results).

Thank you for this comment. Nevertheless, the last table refers to one specific subsection of the results chapter and is hardly collocable as differently as now.

Please separate the discussion and summary into separate chapters.

Appreciating the reviewer’s comment, we have split the discussion (please see Chapter 4, on page 11) and the summary (please see Chapter 5.1, “Conclusions”, on page 12).

I wish you to continue your fruitful work.

Thank you very much; we are confident that the modified manuscript will align with the reviewer’s viewpoint.

Reviewer 3 Report

Comments and Suggestions for Authors

Local food products, especially some autochthonous genotypes of ancient local varieties of fruits and vegetables, have recently attracted attention due to their nutritional value and role in biodiversity conservation. Knowing customers’ sensory expectations and individually perceived monetary values is crucial for proper strategic planning. In this study, consumers’ acceptability for the ancient local pear ‘Angelica’ was estimated using WTP elicitation through “JAR+Hedonic+BDM” tasks. The merits and novelty of this study meet the requirements of Foods, and the study was well designed. Some questions in the manuscript need to be improved.

1.        As only one local ancient pear variety in Italy was used in this study, it is better to include “Italy” in the title to make it more accurate.

2.        “In-store experiments” should be referred to in the abstract.   

3.  L174 ‘Participations has been totally randomised. More specially…excluded from partitipation’ should belong to part ‘2.1 participants’.

4. L216 There is a spelling error in payment in Figure 1. 

5. Sentences in L248-250 should move upper.

6. Sentences in L375-378 should move upper, and be combined with the explanation on sourness and juiciness in L365-369. 

7. Grammarly mistakes should be checked thoroughly and corrected.

Comments on the Quality of English Language

Minor editing of English language required

Author Response

Reviewer #3

Local food products, especially some autochthonous genotypes of ancient local varieties of fruits and vegetables, have recently attracted attention due to their nutritional value and role in biodiversity conservation. Knowing customers’ sensory expectations and individually perceived monetary values is crucial for proper strategic planning. In this study, consumers’ acceptability for the ancient local pear ‘Angelica’ was estimated using WTP elicitation through “JAR+Hedonic+BDM” tasks. The merits and novelty of this study meet the requirements of Foods, and the study was well designed.

            We really appreciate the reviewer’s comment.

Some questions in the manuscript need to be improved.

As only one local ancient pear variety in Italy was used in this study, it is better to include “Italy” in the title to make it more accurate.

Thank you for this suggestion. We updated the title as follows:

Sensory Perception and Willingness to Pay for a Local Ancient Pear Variety: Evidence from In-store Experiments in Italy

“In-store experiments” should be referred to in the abstract.

Appreciating the reviewer’s comment, we have updated the abstract as follows (please see lines 17-20):

One hundred twelve non-expert participants recruited during an in-store experiment evaluated overall liking and JAR attributes and were involved in an in-field experimental auction based on the non-hypothetical Becker-deGroot-Marshak (BDM) mechanism.

L174 ‘Participations has been totally randomised. More specially…excluded from partitipation’ should belong to part ‘2.1 participants’.

Thank you for this suggestion. The following sentence was inserted into part 2.1, on page 4 lines 142-145.

Participation has been randomised. More specifically, only those passers-by who had not participated in previous studies were invited. All people involved in the organisation of the study were excluded from participation.

L216 There is a spelling error in payment in Figure 1.

            Thank you for highlighting the typo. We have fixed the term “payment”.

Sentences in L248-250 should move upper.

Thank you for this suggestion. The following sentence was inserted on page 6, lines 251-255.

SEM is a type of multivariate analysis that combines factor and path analysis. SEM focused exclusively on path analysis because no latent variables are involved in this study. The structural model, expressed in terms of the relationship between variables, was estimated and tested to evaluate the theoretical links.”

Sentences in L375-378 should move upper, and be combined with the explanation on sourness and juiciness in L365-369.

Thank you for this suggestion. Based on other reviewers’ comments, we decided to move this sentence to a separate paragraph of conclusions on page 12, lines 418-421.

Grammarly mistakes should be checked thoroughly and corrected.

            Grammarly mistakes have been checked and fixed.

Reviewer 4 Report

Comments and Suggestions for Authors

- The literature cited is somewhat old. For example, only 6 references are from 2020 onwards. It is suggested to add new literature.

- Correct the English and improve the presentation (use text without symbols) of Figure 1.

- What are the theoretical implications of this research?

- The authors should better explain how they obtained equation 1. Is Log a natural logarithm?

- Is there association between the attributes evaluated (e.g., sweetness with aroma)? That is, it is possible to identify dimensions.

- Figure 5 needs to be explained better. Why are those 2 axes chosen?

- Baron and Kenny's approach is an old approach. The authors could use AMOS and identify the indirect effect in a simpler way.

Author Response

- The literature cited is somewhat old. For example, only 6 references are from 2020 onwards. It is suggested to add new literature.

Appreciating the reviewer’s comment, we have updated the literature in the introduction.

Added Bibliography:

1 Cappelli, L.; D’ascenzo, F.; Ruggieri, R.; Gorelova, I. Is Buying Local Food a Sustainable Practice? A Scoping Review of Consumers’ Preference for Local Food. Sustain. 2022, 14, 1–17, doi:10.3390/su14020772.

13 Roosen, J.; Neubig, C.M.; Staudigel, M.; Agovi, H. Product Appeal, Sensory Perception and Consumer Demand. Eur. Rev. Agric. Econ. 2023, 50, 1338–1363, doi:10.1093/erae/jbad020.

14 Cardona, M.; Izquierdo, D.; Barat, J.M.; Fernández-Segovia, I. Intrinsic and Extrinsic Attributes That Influence Choice of Meat and Meat Products: Techniques Used in Their Identification. Eur. Food Res. Technol. 2023, 249, 2485–2514, doi:10.1007/s00217-023-04301-1.

2 da Costa Marques, S.C.; Mauad, J.R.C.; de Faria Domingues, C.H.; Borges, J.A.R.; da Silva, J.R. The Importance of Local Food Products Attributes in Brazil Consumer’s Preferences. Futur. Foods 2022, 5, 100125, doi:https://doi.org/10.1016/j.fufo.2022.100125.

33 Gracia, A.; Cantín, C.M. Effects of Consumers’ Sensory Attributes Perception on Their Willingness to Pay for Apple Cultivars Grown at Different Altitudes: Are They Different? Foods 2022, 11, doi:10.3390/foods11193022.

34 Boccia, F.; Alvino, L.; Covino, D. This Is Not My Jam: An Italian Choice Experiment on the Influence of Typical Product Attributes on Consumers’ Willingness to Pay. Nutr. Food Sci. 2023, doi:10.1108/NFS-04-2023-0076.

38 Kamal Gosh, U.D.; Dey, M.M. Consumers’ Willingness-to-Pay for Newly Developed U.S. Farm-Raised Convenient Catfish Products: A Consumer-Based Survey Study. Aquac. Econ. \& Manag. 2022, 26, 332–358, doi:10.1080/13657305.2022.2060374.

- Correct the English and improve the presentation (use text without symbols) of Figure 1.

Thank you for this suggestion. English was correct, and symbols were dropped.

- What are the theoretical implications of this research?

Academic/theoretical implications were described in section 5.2.

- The authors should better explain how they obtained equation 1. Is Log a natural logarithm?

Thank you for pointing out this aspect. The log is indeed a natural logarithm. Equation 1 represents a so-called “constant elasticity model” (Wooldridge, pp. 43-44). In other words, the estimate of beta, in this case, is interpreted as (%∆????????)/(%∆?????), that is, approximately an elasticity. In light of the Reviewer’s  suggestion, the text surrounding Eq. 1 has been changed as follows (please see on pages 7-8, lines 299-311:

The empirical demand for pear ‘Angelica’ is analysed using a “constant elasticity model” [47], i.e., a log-log model estimated through OLS as witnessed by Equation 1, where natural logarithms are employed.

log(Quantity)= α +  β x log(Price)

In particular, parameter β, being defined as (%∆ Quantity)/(%∆ Price) [47], clearly represents an elasticity. The relationship comprises 35 observations, i.e., 35 unique prices for 35 pear Angelica quantities. The estimates are  significant. It is worth emphasising that the elasticity of the demand curve concerning price is equal to -1.60. The sign reflects the downward-sloping behaviour typical of demand curves. More precisely, a 1% increase in the pear ‘Angelica’ price reduces the quantity consumed by 1.60%.”

Added Bibliography:

Wooldridge, J.M. Introductory Econometrics: A Modern Approach; 5th Edition; South-Western Cengage Learning, 2013; ISBN 9781111534394.

- Is there association between the attributes evaluated (e.g., sweetness with aroma)? That is, it is possible to identify dimensions.

Thank you for this suggestion, which could be considered in another work specifically centred on sensory analysis. We have considered not exploring the possible association between sweetness and aroma to identify other sensorial dimensions because they are generally treated separately in sensorial studies.

- Figure 5 needs to be explained better. Why are those 2 axes chosen?

According to the literature, penalty analysis considers the mean drop in overall liking (y-axis) and the % of consumers (x-axis). As explained on page 9, lines 322-325, Figure 5 shows the average drops in overall liking as a function of the proportion of consumers that selected the option “out-of-JAR” to describe one of the attributes of the pear sample, thus indicating a need for product improvement.

- Baron and Kenny’s approach is an old approach. The authors could use AMOS and identify the indirect effect in a simpler way.

Thank you for this suggestion, but considering we used STATA, and not SPSS-AMOS, and also STATA identify the indirect effect independently of Baron and Kenny’s approach (please see table 3 as results of STATA analysis of the direct and indirect effects), we have considered not to replicate the findings using SPSS-AMOS.

Round 2

Reviewer 2 Report

Comments and Suggestions for Authors

Dear Editor/ Authors,
I am completely satisfied with the corrections made, regarding my suggestions.
Sincerely,

Author Response

Thank you very much for the Reviewer's comments. 

These comments offer us the possibility of improving the work substantially.

Reviewer 4 Report

Comments and Suggestions for Authors

- The symbol "ln" refers to natural logarithms. “Log” is incorrect.

- Figure 1: Symbols should be replaced with texts.

- In equation 1 obtained, it must be explained how the 35 observations were collected (Price and Quantity).

- Figure 5 needs to be explained better in the document.

Comments on the Quality of English Language

Minor editing of English language required

Author Response

- The symbol “ln” refers to natural logarithms. “Log” is incorrect.

Appreciating the reviewer’s comment, we have changed the symbol.

- Figure 1: Symbols should be replaced with texts.

Appreciating the reviewer’s comment, we have replaced the symbols with texts.

- In equation 1 obtained, it must be explained how the 35 observations were collected (Price and Quantity).

Following the reviewer’s suggestion, it was explained how the 35 observations were defined (please see on page 8, rows 305-306) as follows:

35 observations out of 112 were obtained because multiple equal price values happened.

- Figure 5 needs to be explained better in the document.

Appreciating the reviewer’s comment, we have better described Figure 5 (please see on page 9, lines 325-332) as follows:

Specifically, in the penalty analysis, respondents are grouped in the “Not-To-JAR” (i.e. “Too little” or “Too much”) or JAR group. Then, the percentage of consumers in “Not-To-JAR” is calculated, and corresponding mean liking scores for the “Not-To-JAR” and JAR categories are estimated. Results are thus represented graphically by plotting the penalty (i.e. the mean drops of liking) against the percentage of consumers “Not-To-JAR”. A penalty is usually not considered if the percentage of consumers in the “Not-To-JAR” group is less than 20%.

Minor editing of English language required

English was edited to check for mistakes.
